# Expression of NAC1 Restrains the Memory Formation of CD8^+^ T Cells during Viral Infection

**DOI:** 10.3390/v14081713

**Published:** 2022-08-04

**Authors:** Liqing Wang, Anil Kumar, Jugal Kishore Das, Yijie Ren, Hao-Yun Peng, Darby Jane Ballard, Xiaofang Xiong, Jacob Rance Davis, Xingcong Ren, Jin-Ming Yang, Jianxun Song

**Affiliations:** 1Department of Microbial Pathogenesis and Immunology, Texas A&M University Health Science Center, Bryan, TX 77807, USA; 2Department of Biochemistry and Biophysics, Texas A&M University, College Station, TX 77843, USA; 3Department of Toxicology and Cancer Biology, Department of Pharmacology and Nutritional Science, and Markey Cancer Center, University of Kentucky College of Medicine, Lexington, KY 40536, USA

**Keywords:** NAC1, vaccinia virus, CD8^+^ T cell memory, IRF4

## Abstract

Nucleus accumbens-associated protein 1 (NAC1) is a transcription co-factor that has been shown to possess multiple roles in stem cell and cancer biology. However, little is known about its roles in regulation of the immune system. In the current study, we observed that expression of NAC1 impacted the survival of CD8^+^ T cells in vitro. NAC1^−/−^ CD8^+^ T cells displayed lower metabolism, including reduced glycolysis and oxidative phosphorylation. In vivo, compared with wild-type (WT) mice, NAC1^−/−^ mice produced a lower response to vaccinia virus (VACV) infection, and viral antigen (Ag)-specific CD8^+^ T cells decreased more slowly. Additionally, we observed that the NAC1^−/−^ mice demonstrated a stronger memory formation of viral Ag-specific CD8^+^ T cells post-viral infection. Mechanically, we identified that compared with WT CD8^+^ T cells, the Interferon Regulatory Factor 4 (IRF4), a key transcription factor in T cell development, was highly expressed in NAC1^−/−^ CD8^+^ T cells, insinuating that IRF4 could be a critical regulatory target of NAC1 in the memory formation of CD8^+^ T cells. Our results indicate that NAC1 restrains the memory formation of CD8^+^ T cells by modulating IRF4, and targeting NAC1 may be exploited as a new approach to boosting CD8^+^ T cell memory.

## 1. Introduction

Memory T cells are formed by the host in the process of eliminating invading pathogens [1]. Upon repeated infection by the same pathogen, these memory T cells are able to respond quickly to provide protective immunity [2]. This form of immunologic memory is vital for raising an immune response against many infectious agents such as viruses and bacteria [2,3]. Memory CD8^+^ T cells play a critical role during acute or chronic viral infection [4] and improving and prolonging CD8^+^ T cell memory could help strengthen the protective efficacy of vaccine design strategies and boost immune responses. Additionally, it has been appreciated that the memory T-cell-based immunotherapy has better efficacy than the effector T-cell-based immunotherapy in cancer treatments [5]. Hence, the strategy to improve CD8^+^ T cell memory formation may provide effective prevention of virus reinfection and improve the efficacy of T-cell-based immunotherapy. A number of co-stimulatory signals such as CD28, OX40, and transcriptional factors, such as Forkhead Box O1 (FoxO1) and T Cell Factor 1 (TCF1), have been shown to influence T cell memory formation [6,7,8,9].

Nucleus accumbens-associated protein 1 (NAC1) is a transcriptional co-repressor [10] and plays multiple important roles in stem cell and cancer biology. For instance, NAC1 is critical for the pluripotency maintenance of embryonic stem cells through its interaction with cell reprogramming factors (Sox2, Oct4, and c-Myc) [11]. NAC1 contributes to chemo-resistance by regulating autophagy through High Mobility Growth Box1 (HMGB1) [12]. Additionally, NAC1 inhibits cellular senescence [13] and inactivates tumor suppressor genes [14]. Furthermore, NAC1 boosts tumor cell metabolism through the NAC1-HDAC4-HIF1α pathway [15]. Recently, we demonstrated that NAC1 modulates the functional activity of regulatory T cells (Tregs) [16]. In this study, we investigated the role of NAC1 in the regulation of T cell memory formation using wild type (WT) and NAC1-deficient (^−/−^) mice. We found that NAC1 has important roles in regulating CD8^+^ T cell function, survival, and memory. We also found that Interferon Regulatory Factor (IRF4), a transcription factor that is closely associated with T cell receptor (TCR) signaling [17,18], is involved in this regulation.

## 2. Materials and Methods

### 2.1. Animal Experiments

Eight- to twelve-week-old C57BL/6 (B6) mice were purchased from The Jackson Laboratory (Bar Harbor, ME). NAC1^−/−^ mice (B6 background) are bred and maintained in the Texas A&M University Laboratory Animal Resources and Research Facility. All animal studies were conducted in accordance with the guidelines of the Institutional Animal Care and Use Committee (IACUC #2018-0065), Texas A&M University.

### 2.2. T Cell Isolation, T Cell Activation, and T Cell Culture

CD8^+^ T cells were isolated from mouse spleen and lymph nodes (LNs) using MojoSort Mouse CD8^+^ Naive T Cell Isolation Kit (BioLegend #480044). T cells were activated by plate-coated 4 μg/mL anti-CD3 (clone 2C11) and 4 μg/mL anti-CD28 (clone 37.51) antibodies (Abs) in T cell culture medium. T cells were cultured in RPMI medium with 10% FBS, 1% NEAA, 55 μM 2-ME, 2 mM L-glutamine, and 1% Penicillin-streptomycin. T cells were split based on their density.

### 2.3. Virus Preparation and Titration

Vaccinia virus Western Reserve strain (VACV-WR) stock was prepared and grown in HeLa cells. When the HeLa cells neared confluency, they were infected at the optimal multiplicity of infection (MOI) around 2 PFU/cell. Later, the vaccinia virus stock was titrated with Vero C1008 cells through a plaque assay. When the Vero cells reached confluence, the virus stock was serial diluted and added to each 6-well. After two days of incubation, the plaque numbers were counted after Crystal Violet Staining. The detailed methods for both are in the protocol described previously [19]. The viral stock was then kept at −80 °C for future usage.

### 2.4. Viral Infection

VACV-WR infection was performed by intraperitoneal injection (2 × 10^6^ PFU/mouse) as described previously [20].

### 2.5. Western Blot

T cell protein was extracted with M-PER™ Mammalian Protein Extraction Reagent (Thermo Scientific #78503). The protein samples were then collected and quantified using BCA protein assay (Thermo Fisher #23225). The IRF4 primary Ab used is rabbit anti-mouse IRF4 Ab (CST #62834T). The β-ACTIN primary Ab utilized is rat anti-mouse ACTIN Ab (BioLegend #664802). The second HRP-conjugated anti-rabbit (BioLegend #406401) and anti-rat (BioLegend #405405) Abs were purchased from BioLegend.

### 2.6. Memory T Cell Tetramer Staining and Flow Cytometry

Pooled superficial cervical, axillary, brachial, and inguinal lymph nodes were combined with the spleen of each mouse for analysis. The tissues were pulverized, and cells were filtered using a 40 μm cell strainer. T cell preparation and staining with different surface markers were described in a previous publication [7]. Following this, the VACV tetramer was used to detect VACV-specific T cells at room temperature for 30 min in a cell-staining buffer (BioLegend #420201). The MHC class I B8R (TSYKFESV) tetramer was synthesized in the NIH Tetramer Core. All flow cytometry experiments were completed in the Texas A&M University COM-CAF core facility with the BD Fortessa X-20. The final plotting was performed in FlowJo Software.

### 2.7. RNA Extraction, cDNA Synthesis, and qPCR

RNA extraction was completed with the RNeasy Mini Kit (Qiagen #74104). DNA was removed by TURBO DNA-free Kit (Ambion #AM1907). The cDNA was synthesized with High-capacity cDNA Reverse Transcription Kit (Thermo Fisher #4368813). The qPCR was accomplished with primers described below. *Irf4*: Forward- GCAATGGGAAACTCCGACAGT,

Reverse- CAGCGTCCTCCTCACGATTGT [18].

*Gapdh*: Forward-GTTGTCTCCTGCGACTTCA, Reverse-GGTGGTCCAGGGTTTCTTA.

Bio-Rad CFX384 Touch Real-Time PCR Detection System was used to perform qPCR.

### 2.8. Seahorse Assay

The Seahorse assays were performed with Agilent Seahorse XF Cell Mito Stress Kit (#103010-100) and Agilent Seahorse XF Glycolytic Rate Assay Kit (#103346-100) according to their user guides. Approximately 2 × 10^5^ cells were plated in each well of the microplate before the glycolytic rate and mitochondrial stress tests. The drugs were injected into each sample at different times. The Extra Cellular Acidification Rate (ECAR) was measured in the glycolytic rate and the Oxygen Consumption Rate (OCR) was tested to indicate oxidative phosphorylation.

### 2.9. CHIP-Seq

The CHIP-seq sample preparation was finished with Zymo-Spin CHIP Kit (#D5209). The Ab utilized for NAC1 was the mouse NAC1 Ab (BioLegend #849301). The IgG control Ab selected was Go-ChIP-Grade™ Purified Mouse IgG1 (BioLegend #401409). The sample was sequenced in TIGSS Molecular Genomics Core at Texas A&M University. The sequencing data were then visualized by the Integrative Genomics Viewer (IGV).

### 2.10. CFSE Labeling

CD8^+^ T cells were isolated from pooled LNs and spleen. Then, the T cells were labeled with CFSE for 10 min at room temperature. Then, cells were activated with precoated anti-CD3 and soluble anti-CD28 Abs as we described in Section 2.2. After two days, the samples were analyzed by flow cytometry.

### 2.11. Statistical Analysis

In the statistical analysis, we used Student’s *t*-test. *p*-values lower than 0.05 were considered significant.

## 3. Results

### 3.1. Generation of VACV for Mouse Infection

We used the vaccinia virus (VACV) as the tool for experiments on T cell memory formation because VACV can create a relatively strong and long-lasting T cell memory [21]. HeLa cells were infected with VACV according to the VACV stock preparation protocol [19]. We observed that the HeLa cells altered their morphology 2 days post-infection (Figure 1A). HeLa cells shrank in size and became more spherical, resulting in a less attached status. After the collection of VACV stock, we used a plaque assay to quantify the VACV titer. Following serial dilutions, we found that the 10^−7^ dilution provided a reliable plaque number (Figure 1B). Quantification of the plaque assays showed that the virus titer was 3.85 × 10^8^ PFU/mL (Figure 1B).

### 3.2. Defects in the Survival of NAC1^−/−^ CD8^+^ T Cells

NAC1 has been shown to regulate cancer cell survival [12,13,15]. Here, we investigated whether NAC1 could interfere with T cell proliferation and survival. We first compared cell proliferation between CD8^+^ T cells from WT and NAC1^−/−^ mice. Naive CD8^+^ T cells from the pooled LNs and spleen were labeled with carboxyfluorescein succinimidyl ester (CFSE) and stimulated with plate-coated anti-CD3 plus soluble anti-CD28 Abs, and then cell proliferation was determined by CFSE dilution. We observed that NAC1^−/−^ CD8^+^ T cells almost retained similar proliferation compared with the WT CD8^+^ T cells (Figure 1C). We next compared cell survival between CD8^+^ T cells from WT and NAC1^−/−^ mice. Three days after activation, NAC1^−/−^ CD8^+^ T cells doubled their population, but WT CD8^+^ T cells showed an almost 3-fold increase. On day 4, NAC1^−/−^ CD8^+^ T cells decreased their population, whereas WT T cells had an 8-fold increase in cell number compared with that on day 0. Moreover, 4 days later, WT CD8^+^ T cells also maintained robust survival as compared with NAC1^−/−^ CD8^+^ T cells (Figure 1D). These results indicate that loss of NAC1 negatively affects the survival of CD8^+^ T cells.

### 3.3. Defects in Glycolysis and Oxidative Phosphorylation Rate of NAC1^−/−^ CD8^+^ T Cells

As NAC1 can regulate tumor cellular metabolism [15], we hypothesized that this transcription co-regulator also plays a role in T cell metabolism. To test this hypothesis, we used Agilent Seahorse Assay to analyze glycolysis and oxidative phosphorylation in T cells. Following the addition of rotenone and antimycin A (Rot/AA), we observed that NAC1^−/−^ CD8^+^ T cells had lower ECAR than WT cells (Figure 2A,C), indicating that NAC1^−/−^ CD8^+^ T cells have a reduced glycolytic rate. After adding carbonyl cyanide-p-trifluoromethoxyphenylhydrazone (FCCP), a potent uncoupler of mitochondrial oxidative phosphorylation which disrupts ATP synthesis by transporting protons across cell membranes, NAC1^−/−^ CD8^+^ T cells had lower OCR than WT cells (Figure 2B,D). These results suggest that after activation, NAC1^−/−^ CD8^+^ T cells have a reduced metabolic rate likely due to metabolic reprogramming caused by loss of NAC1.

### 3.4. Sustained Survival of the VACV-Specific CD8^+^ T Cell in NAC1^−/−^ Mice

We next assessed how NAC1 influences antigen (Ag)-specific cell generation and survival in vivo. To monitor the Ag-specific T cell proliferation and survival in vivo, we challenged the mice with VACV (2 × 10^6^ PFU/mouse). The number and frequency of the VACV-specific CD8^+^ T cells were determined in the next 5 weeks. We observed that both WT and NAC1^−/−^ mice responded to VACV infection (Figure 3A,B). The T cell responses peaked on day 7 after the VACV challenge, then started to decline. Levels of IFNγ secretion were higher in WT CD8^+^ B8R^+^ T cells (Appendix A), but no obvious difference was observed for TNFα cytokine secretion (Appendix A). The VACV-specific CD8^+^ T cell number was significantly higher in WT than in NAC1^−/−^ mice during the initial 3 weeks (Figure 4A). The VACV-specific CD8^+^ T cell numbers in both groups were decreased to a similar level on day 35 after viral infection (Figure 4A). Although NAC1^−/−^ mice revealed a lower number of VACV-specific CD8^+^ T cells, its cell frequency in these mice was not continually lower than that in the WT group (Figure 4B). In fact, after one week, the frequency of the VACV-specific CD8^+^ T cells decreased more slowly in the NAC1^−/−^ mice than that in the WT controls. In contrast, on day 21, NAC1^−/−^ mice maintained a higher frequency of the VACV-specific CD8^+^ T cells than WT controls. There was no significant difference in the VACV-specific CD8^+^ T cell frequency between those two groups 35 days later. Our results suggest that NAC1 negatively affects the survival of the Ag-specific CD8^+^ T cell in vivo.

### 3.5. Enhanced CD8^+^ T Cell Memory Formation in NAC1^−/−^ Mice

When a host eliminates a viral infection, memory T cells maintain homeostatic proliferation and survive past the initial immune response [1]. To determine the effect of NAC1 on memory of CD8^+^ T cells, we used VACV to challenge mice and 35 days later, examined the memory formation of T cells in both WT and NAC1^−/−^ mice. We observed a higher frequency of the VACV-specific memory CD8 T cells in NAC1^−/−^ mice than that in WT controls, as analyzed by flow cytometry (Figure 5B and Figure 6B). Notably, there was no significant difference in the number of the VACV-specific memory CD8^+^ T cells between WT and NAC1^−/−^ groups (Figure 6A), which was consistent with the Ag-specific CD8^+^ T cell number shown in Figure 4A. Although the NAC1^−/−^ mice were less responsive to the VACV challenge and generated a smaller number of the VACV-specific CD8^+^ T cells in the early stage, their slowly decreased VACV-specific T cells resulted from the longer and better memory T cell formation after the effector stage. We further analyzed the memory T cell subsets (Figure 5C) and found that a small portion of the tissue-resident memory (CD44^hi^CD69^hi^ CD197^low^) and central memory (CD44^hi^CD69^low^ CD197^hi^) CD8^+^ T cells existed, and most of them were effector memory (CD44^hi^CD69^low^ CD197^low^) T cells. There was no significant difference in the number of the tissue-resident memory T cells between NAC1^−/−^ and WT mice (Figure 6C). On day 7 after VACV challenge, we also discovered that CD8^+^ B8R^+^ CD127^+^ CD62L^+^(precursor) cells were higher in NAC1^−/−^ mice (Appendix A). For the terminally differentiated CD8^+^ B8R^+^ CD127^−^ KLRG1^+^ cells, that there was no significant difference relative to the wild-type (Appendix A). Based on these data, it seems that those higher memory precursor cells in NAC1^−/−^ mice are related to later higher memory T cell frequency. These results demonstrate that NAC1 suppresses memory CD8^+^ T cell formation in vivo.

### 3.6. Involvement of IRF4 in the NAC1-Mediated Restrain of CD8^+^ T Memory Formation

It was reported that IRF4 can support the resident memory CD8^+^ T cell maintenance [18] and play a pivotal role in T cell activation [17]. Therefore, we examined the IRF4 expression in CD8^+^ T cells. Naive CD8^+^ T cells were isolated from WT or NAC1^−/−^ mice and stimulated with plate-coated anti-CD3 plus soluble anti-CD28 Abs, and IRF4 protein expression was determined by Western blot. We found that before activation, the basal expression of IRF4 was barely detectable in both groups, but evidently increased after activation. Notably, on day 3, NAC1^−/−^ T cells had a much higher expression of IRF4 than the WT control (Figure 7A), suggesting that NAC1 may control the expression of IRF4. However, there was no significant difference in the mRNA level for *Irf4* between the two groups (Figure 7B). In addition, the CHIP-seq analysis did not reveal any binding of NAC1 protein to *Irf4* (Figure 7C). Thus, the regulation of IRF4 by NAC1 does not occur at transcriptional level but at a post-transcriptional level.

## 4. Discussion and Conclusions

In this study, we used a mouse system to investigate the regulation of CD8^+^ T cells by the NAC1 during viral infection. We showed that NAC1 controlled CD8^+^ T cell survival (Figure 1) and regulated cell metabolism (Figure 2). Using VACV, we demonstrated that during the virus infection, loss of NAC1 sustained the survival of Ag-specific CD8^+^ T cells (Figure 4). At the recovery stage, NAC1^−/−^ CD8^+^ T cells maintained higher memory T cell formation (Figure 6) and IFR4 expression (Figure 7) than WT cells. These results indicate that NAC1 can repress CD8^+^ T cell memory formation and IRF4 is involved in this regulation.

It has been reported that NAC1 is important for tumor growth and metabolic reprogramming; consequently, targeting NAC1 can suppress tumor growth [12,15]. For example, the NAC1 inhibitor, NIC3, can interrupt NAC1 homodimerization and shows antitumor activity [22]. However, little is known about the role of NAC1 in immune cells. The role of NAC1 in T cell biology may be multifaceted, potentially playing different roles in different T cell subsets such as CD8, Th1, Th2, Th17, and Tregs. Recently, we reported the negative regulation of Tregs by NAC1 [16]. In this study, we found that loss of NAC1 impaired CD8^+^ T cell survival after activation in vitro and NAC1^−/−^ CD8^+^ T cells demonstrated decreased glycolysis after activation. Because T cells require different metabolic profiles during different stages of differentiation [23], NAC1-mediated alternations in glycolysis and oxidative phosphorylation may result in different T cell memory statuses during pathogen infections.

There are three important phases during T cell anti-viral immune response: activation and proliferation, death, and memory formation [4]. NAC1 may differentially influence T cells during distinct phases after viral infection. In this study, we show that NAC1 supported T cell survival. After viral infection, NAC1^−/−^ mice developed a smaller number of virus-specific CD8^+^ T cells. However, these virus-specific CD8^+^ T cells died slower than WT controls after the effector peak. After 35 days, NAC1^−/−^ mice maintained a higher frequency of virus-specific memory CD8^+^ T cells. These results indicate that NAC1 represses CD8^+^ T cell memory formation and that loss of NAC1 reduces the death of memory CD8^+^ T cells. Therefore, targeting NAC1 may be explored as an effective approach to improving the effectiveness of some vaccines whose protection period is short. For example, the effectiveness of the COVID-19 vaccine had been proved to gradually decrease after 5–6 months among fully vaccinated people [24,25]. Thus, it is advised that a booster should be given after 6 months. Alternatively, we may slow down the decrease in protection by prolonging the lifetime and population of memory T cells to maintain vaccine effectiveness.

In the setting of chronic infection, exhausted T cells express higher IRF4 and repress memory T cell formation [26]. It has been reported that IRF4^−/−^ mice develop fewer memory CD8^+^ T cells due to their initially poor activation [27]. Furthermore, in the conditional tamoxifen-induced Cre-lox knockout system, it was found that IRF4 supported resident memory T cell formation [18]. Therefore, the roles of IRF4 in T cell memory formation are still controversial [17,18,26,27]. Our results show that NAC1 restrained overall CD8^+^ T cell memory formation but not the tissue-resident memory T cell population, and this was accompanied by down-regulation of IRF4 protein. Down-regulation of IRF4 appears to occur post-transcription, as our PCR analysis did not show significant difference in *irf4* mRNA between WT and NAC1^−/−^ CD8^+^ T cells (Figure 7). However, the precise mechanism by which NAC1 modulates IRF4 expression remains to be elucidated.

At the same time, another limitation of our work is the usage of whole NAC1 knockout mice models during VACV infection. However, we consider the NAC1 effect to be intrinsic based on the in vivo experiments. In the future, we could try either a conditional knockout mice model or an adoptive transfer experiment to improve the experiment design.

Taken together, this study demonstrates the critical regulation of CD8^+^ T cell memory formation by NAC1, likely through IRF4, and suggests that targeting NAC1 may provide a potentially effective approach to enhancing vaccine efficacy.

## Figures and Tables

**Figure 1 viruses-14-01713-f001:**
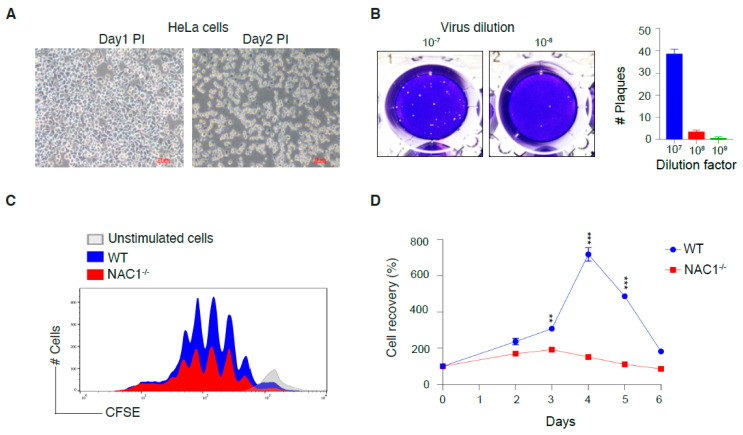
Preparation of Vaccinia virus Western Reserve strain (VACV-WR) stock and defects in the survival of NAC1^−/−^ CD8^+^ T cells. (**A**) Confluent plates of HeLa cells were infected at an optimal multiplicity of infection (MOI) of 2 PFU/cell. Cellular morphology was monitored during the first two days post-infection. (**B**) When the HeLa cells reached 100% confluence, the virus stock was serially diluted and added to each 6-well. After two days of incubation, the plaque numbers were counted after crystal violet staining. The viral plaque number was reliable at the dilution factor 10^7^. (**C**) CD8^+^ T cells were isolated from WT and NAC1^−/−^ mice, then further stained with CFSE and cultured for 2 days before flow-cytometric analysis. (**D**) Isolated T cells were seeded into 48-well plates and split at confluence. The live cell number was counted with trypan blue staining. Repeat *n* = 4. The *p*-values in the figure are 0.0043 (day 3), 1.5 × 10^−5^ (day 4), and 1.4 × 10^−6^ (day 5). (**, *p* < 0.01; ***, *p* < 0.001).

**Figure 2 viruses-14-01713-f002:**
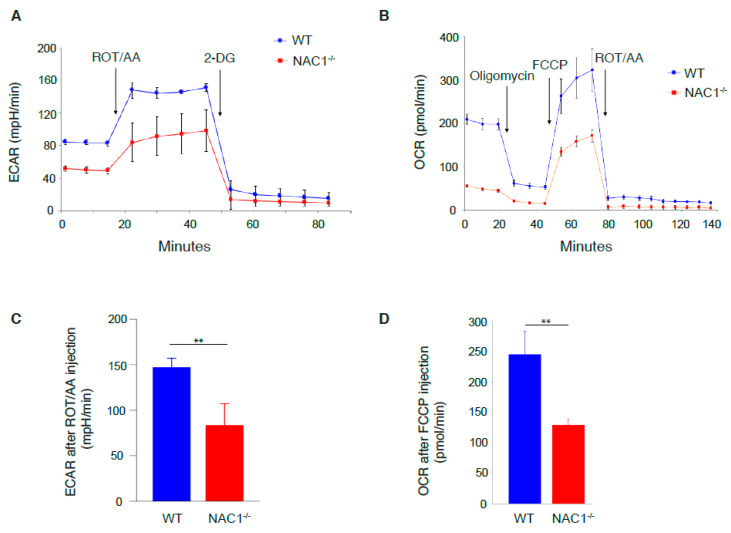
Defects in glycolysis and oxidative phosphorylation rate of NAC1^−/−^ CD8^+^ T cells. Cell metabolism was tested using the Agilent Seahorse Assay on day 3 after activation. (**A**,**C**) The cells were activated with coated anti-CD3 and soluble anti-CD28 Abs. Then, 2 × 10^5^ cells were plated in each well of seahorse microplate for glycolytic rate testing. The drugs were injected into each well at the time indicated in the figures. The Extra Cellular Acidification Rate (ECAR) was measured as a proxy for glycolytic rate. The ECAR level after ROT/AA injection was compared between WT and NAC1^−/−^ groups. Sample numbers (*n* = 4), the *p*-value is 0.0076 (**B**,**D**). The activated 2 × 10^5^ T cells were seeded in a microplate before mitochondrial stress testing. The different drugs were added to the well according to the indicated time points. The Oxygen Consumption Rate (OCR) was tested as a proxy for oxidative phosphorylation. The OCR level was then compared after FCCP injection between the two groups. Sample numbers (*n* = 4), *p*-value is 0.0054 (**, *p* < 0.01).

**Figure 3 viruses-14-01713-f003:**
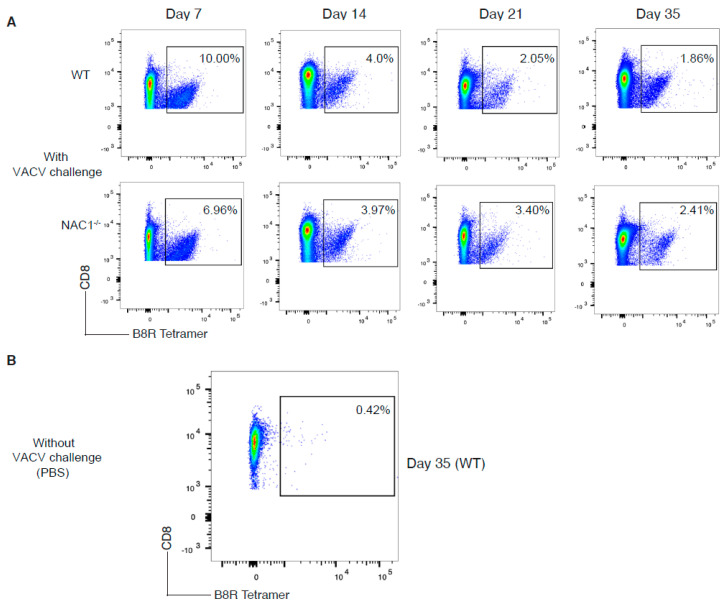
Comparison of development of viral Ag-specific CD8^+^ T cells. The VACV-specific CD8^+^ T cell frequency was monitored in WT or NAC1^−/−^ mice for 35 days after VACV challenge. (**A**) Mice were challenged with VACV at 2 × 10^6^ PFU/mouse. The spleen and LNs (superficial cervical, axillary, brachial, and inguinal nodes) were dissected, smashed, and stained with CD8 Ab and B8R tetramer. On day 7, day 14, day 21, and day 35, Ag-specific cell frequencies were analyzed by flow cytometry. (**B**) VACV-uninfected mice were used as a control. The spleen and lymph nodes were also dissected, smashed, and stained with CD8 Ab and B8R tetramer. The plotting data shown are representative of three identical experiments (*n* = 5).

**Figure 4 viruses-14-01713-f004:**
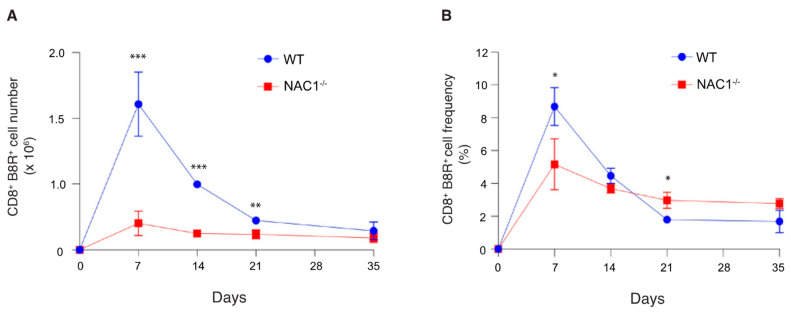
Sustained VACV-specific CD8^+^ T cell survival in NAC1^−/−^ mice. The VACV-specific CD8^+^ T cell number and cell frequency changed within 35 days post-infection. (**A**) Mice were challenged with VACV at 2 × 10^6^ PFU/mouse. The spleen and lymph nodes (superficial cervical, axillary, brachial, and inguinal nodes) were dissected and smashed. The total live cell number for each mouse was calculated with trypan blue staining using a Bio-Rad cell counter. T cells were stained with CD8 Ab and B8R tetramer and analyzed by flow cytometry. Total Ag-specific CD8^+^ T cells were calculated for each mouse (*n* = 5). The *p*-values are 0.00038 (day7), 8.2 × 10^−5^ (day 14), and 0.0066 (day 21). (**B**) The Ag-specific CD8^+^ T cell frequency was monitored for 35 days. P-values are 0.011 (day 7), and 0.016 (day 21). (*, *p* < 0.05; **, *p* < 0.01; ***, *p* < 0.001).

**Figure 5 viruses-14-01713-f005:**
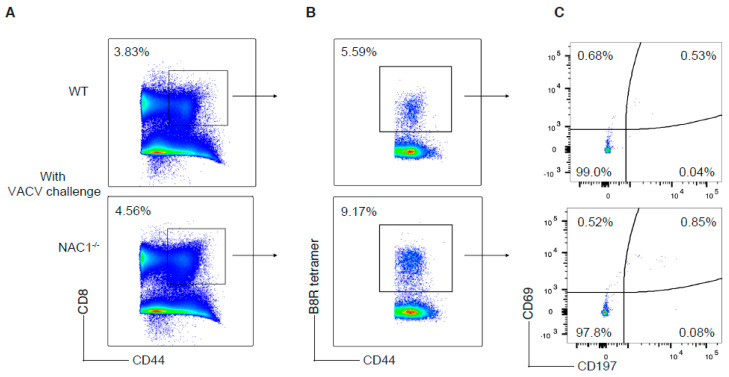
Analyses of CD8^+^ T cell memory formation. T cell memory population was investigated in WT or NAC1^−/−^ mice after VACV challenge. The mice were sacrificed on day 35. The spleen and LNs were dissected, smashed, and stained before flow cytometry. (**A**) Memory CD8^+^ T cells (CD8^+^ CD44^+^). (**B**) VACV-specific memory CD8^+^ T cells (CD8^+^ CD44^+^ B8R^+^), gating on memory CD8^+^ T cells. (**C**) Tissue-resident memory (CD44^hi^CD69^hi^ CD197^low^), central memory (CD44^hi^CD69^low^ CD197^hi^) and effector memory (CD44^hi^CD69^low^ CD197^low^) T cells, gating on VACV-specific memory CD8^+^ T cells. The plotting data shown are representative of three identical experiments (*n* = 5).

**Figure 6 viruses-14-01713-f006:**
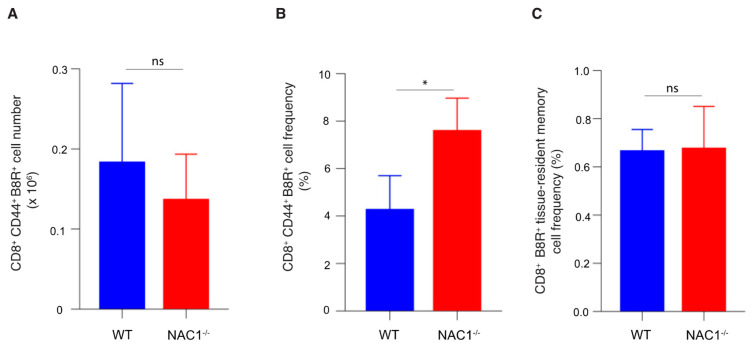
Enhanced CD8^+^ T cell memory formation in NAC1^−/−^ mice. Quantification for T cell memory population after 35 days. The mice were sacrificed on day 35. The spleen and lymph nodes were dissected and smashed. The total live cell number for each mouse was calculated with trypan blue staining through a Bio-Rad cell counter. T cell surface markers were stained before flow cytometry. (**A**) The VACV-specific memory CD8^+^ T cell (CD8^+^ CD44^+^ B8R^+^) number for each mouse had been calculated and compared between WT and NAC1^−/−^ groups. (**B**) The VACV-specific memory CD8^+^ T cell (CD8^+^ CD44^+^ B8R^+^) frequency for each mouse was collected and analyzed. The *p*-value is 0.040. (**C**) Tissue-resident memory (CD44^hi^CD69^hi^ CD197^low^) T cell frequency had been analyzed between WT and NAC1^−/−^ groups. (*, *p* < 0.05; ns, *p* > 0.05).

**Figure 7 viruses-14-01713-f007:**
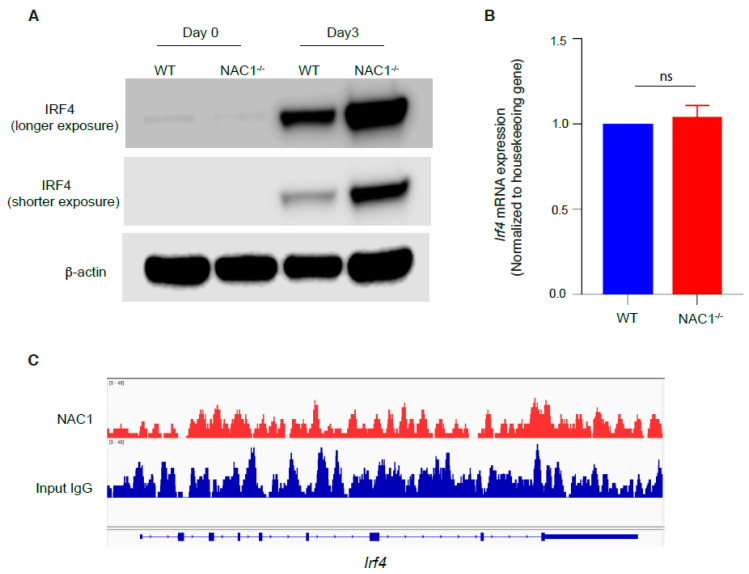
Regulation of IRF4 in CD8^+^ T cells by NAC1. CD8^+^ T cells were isolated from WT or NAC1^−/−^ mice and analyzed for expression of IRF4. (**A**) Protein expression by Western blot. For the day 0 sample, T cells were not activated. For the day 3 sample, T cells were activated and cultured for 3 days. (**B**) mRNA expression by Q-PCR. CD8^+^ T cells had been cultured for 3 days, and then RNA was extracted, and qPCR was performed (ns, *p* > 0.05). (**C**) CHIP-seq analysis. WT CD8^+^ T cells were isolated from mice and CHIP was performed with anti-NAC1 and anti-IgG Abs (Input control). The sequenced data was visualized by IGV.

## Data Availability

The data presented in this study are available on request from the corresponding author.

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
