# Peer review of "Expression of NAC1 Restrains the Memory Formation of CD8+ T Cells during Viral Infection"

_viruses, 2022, doi:10.3390/v14081713_

Round 1

Reviewer 1 Report

The authors have provided thorough response to this reviewer's comments. All the changes made by the author are sufficient and acceptable to this reviewer.

Reviewer 2 Report

Thank you for revising the manuscript, which has significantly improved the representation of the work.

This manuscript is a resubmission of an earlier submission. The following is a list of the peer review reports and author responses from that submission.

Round 1

Reviewer 1 Report

To the authors,

In this paper, Liqing Wang et al., from Jim Song’s group examine the role of transcriptional co-repressor NAC1 in CD8 T cells. Briefly, the authors perform in vitro assays for T cell activation and in vivo experiment involving Vaccinia virus (VACV) infection to show that NAC1 plays important role in proliferation and survival of activated/ antigen specific CD8 T cells. In longitudinal studies following VACV infection, authors observe a defect in virus specific CD8 T cells during effector (expansion) phase of antiviral CD8 T cell response in mice lacking NAC1. However, virus-specific memory CD8 T cell frequency is higher in NAC1 deficient mice compared to WT mice. Authors also show upregulation of IRF4 in NAC1 deficient CD8 T cells at protein level but not mRNA level suggesting a possibility of post-transcriptional regulation.

 This is a well written paper and nicely done study with appropriate controls and logical experimental design. It is easy-to-follow and adds new finding related to the role of NAC1 in immune responses. Overall, this is an interesting finding with respect to the role of in NAC1 in viral infection model as it reveals an important role for this transcriptional co-repressor in regulation of antiviral CD8 T cell responses.

I have only few major and few minor comments for authors to consider with a hope to improve the manuscript. If authors decide to modify the text and add explanation for the following points in the discussion section rather than doing lengthy experiments, it will be acceptable to this reviewer.

Major comments:

  1. Did authors see any change in morbidity or mortality of NAC1 deficient mice compared to WT mice? Related to this, were there differences in viral burden? One can expect NAC1 deficient mice to have higher viral burden if they generate significantly less virus-specific CD8 T cells (if CD8 T cells contribute to host resistance in VACV i.p route model?). Have authors considered doing intranasal VACV infection (which is CD8 T cell mediated protection model)?
  2. Have authors examined the expression of NAC1 in CD8 T cells (B8R+) at different days post-VACV infection? Is it upregulated or has transient expression upon activation in CD8 T cells?
  3. Since authors observed defect in expansion phase but increase during contraction or memory formation, is it possible that virus specific CD8 T cells are skewed more towards memory precursors and less towards effector cells (terminally differentiated)? Did authors examine KLRG1 Vs. CD127 expression on B8R positive cells at day 7? This will help to determine if differentiation is altered.
  4. Have authors looked at functionality of virus-specific CD8 T cells in NAC1 deficient mice compared to WT? Are CD8 T cells in NAC1 deficient mice defective in interferon gamma and/ or TNF alpha expression upon B8R peptide stimulation?

Minor comments:

  1. What is the status of NAC1-/- mice at baseline? Do naïve NAC1-/- mice have similar numbers/percentages of T cells compared to WT? Did authors notice any other abnormalities in the naive NAC1-/- mice?
  2. In Figure 7, WB data, the beta actin band at day 3 in NAC1 deficient mice seems stronger (not just IRF4) compared to WT. Any possible explanation for this?
  3. Authors measure resident memory T cells or TRM (with the help of markers CD69) in the lymphoid tissues. But TRMs are usually enriched in peripheral tissues than spleen or LNs. Did authors check TRM cells in peripheral sites such as the lungs? It is also better to add CD103 staining along with CD69 as surface marker determining TRM cells (or do IV exclusion).
  4. One limitation of this study is the use of whole NAC1 mice model during VACV infection. It is unclear if NAC1 effect if T cell intrinsic or extrinsic. Can authors point to this in their discussion section and whether they think it is intrinsic or extrinsic of CD8 T cells?
  5. The heading of the results section 2.4 should be VACV and not VACA
  6. In Figure 3 and 4, when authors show B8R response. Is it spleen or LNs data that is depicted? Authors mention both LNs and spleens were harvested in figure legends, but show plot for one tissue or did they combine? Also, it will be good to specify in figure legends which LN did authors dissect?

Overall, this is a nicely done study and will add value to the literature on the role of NAC1 in immune response. With some improvement, this article will be welcome addition to scientific literature. I wish the authors the best.

Reviewer 2 Report

In their manuscript entitled „Expression of NAC1 restrains the memory formation of CD8+ T cells during viral infection“, the authors investigate the role of NAC1 in the anti-viral CD8+ T cell response during an acute infection and propose a novel role for NAC1 in regulating memory T cell development. Interestingly, the authors find that NAC1-/- CD8+ T cells displayed reduced glycolysis and oxidative phosphorylation and that NAC1 regulated the expression of IRF4, a transcription factor that has previously been linked to regulating the effector vs. memory fate decision in CD8+ T cells. The results are intriguing, but the conclusions drawn from the data presented in this manuscript appear very preliminary. In particular, the mechanistic part of the study is severely underdeveloped. Specifically the following points need to be addressed:

Major points:

-       Based on the data shown in Fig. 1c, the authors conclude that “NAC1-/- CD8+ T cells had substantially reduced proliferation compared with the WT CD8+ T cells.” However, the CFSE dilution profiles of both groups indicate equal numbers of cell divisions in both groups (up to ~5 divisions). Are the differences in CFSE dilutions significant between both groups? Could the authors provide a second readout for T cell proliferation e.g. BrdU pulsing in vivo? This would at least clarify whether defective proliferation of NAC1-/- has a dominant role in shaping the CD8+ T cell response magnitude.

-       What is the composition of the antigen-specific CD8+ T cell population at the peak of the response (e.g. day 8 p.i.). Are there differences in the abundance (absolute cell numbers) of central memory (CD127+CD62L+), effector memory (CD127+CD62L-) T cells and short-lived effector cells (CD127-KLRG1+)?

-       How do recall responses differ between NAC1-/- and WT mice? If memory formation is enhanced for NAC1-/- cells, as proposed by the authors, NAC1-/- CD8+ T cell should also mount superior recall responses.

-       What is the role of NAC1 in thymocyte development i.e. is the naïve T cell compartment between NAC1-/- and WT mice comparable? What is the role of NAC1 for CD4+ T cell development and is this compartment also affected, which might play a role for these experiments? A suitable experiment would be to adoptively transfer equal cell numbers of NAC1-/- or WT Vaccina-specific naïve CD8+ T cells into congenic-mismatched wildtype mice to corroborate the results.

-       The authors state that “NAC1 suppresses memory CD8+ T cell formation in vivo” mainly based on the relative frequencies of antigen-specific cells shown in Fig. 6B. However, absolute cell numbers between NAC1-/- and WT cells are completely identical (Fig. 4A).  This point is central to the manuscript and is not sufficiently addressed. Thus, based on their experimental data, the authors cannot support their claim for a role of NAC1 in memory CD8+ T cell generation.

-       How is the expression of IRF4 regulated by NAC1 at the post-transcriptional level? The authors should further confirm their IRF4 phenotype in their in vivo setting, since IRF4 staining antibodies are readily available.

Minor points:

-       Language needs revising.

Reviewer 3 Report

The manuscript presents work that may provide interesting insights in the expression of NAC1 on the formation of memory T cells in response to vaccination. As far as I can see, the manuscript in its current form has major issues that should be addressed to have it accepted for publication. The data as presented do not sufficiently support some of the main conclusions, and the methods section has major shortcomings that must be fixed to allow interpretation of the data by the reader. These concerns are the following:

1. The abstract concludes that modulation of IRF4 is the mechanism by which NAC1 restrains memory formation of CD8+ T cells. Although the data show that IRF4 is involved, the data do not show strong evidence that modulation of IRF4 is indeed mechanistically involved as an effector molecule in NAC1 modulating the memory formation. Stronger evidence should be provided, or the conclusion should be more nuanced.

2. Figure 1C does not reflect defective proliferation of NAC1-/- T cells that is claimed by the authors. Figure 1C seems to show a lower number of cells being analyzed by flow cytometry. Moreover, multiple experiments must be shown, including statistical testing, like in figure 1D. These are now lacking.

3. Number of repeated measures/samples (n=?) and the statistical tests that resulted in the given p-values are lacking and must be indicated for figure 1, 2, 4, 6.

4. What do the authors mean by 'recovery' of T cells in their in vitro assays? This is a major part of the manuscript's conclusion and suggests that after activation and proliferative expansion, cells would survive better in order to yield a higher number of live memory T cells. Expansion and contraction of memory cells is the sum of proliferation and cells that die upon response. The results show proliferation and the number of cells at a given time point after stimulation. However, the authors did not show analyses of cell death in their assays, whereas information on how many cells died during the culture would provide important information on survival of T cells throughout the assay. 

5. Figure 2: How long had the T cells been cultured with anti-CD3/CD28 before the cells were analysed through Agilent Seahorse Assay?

6. Why is the virus VACV also described as VACA at some points in the manuscript?

7. Figure 3: Did the authors use proper compensation controls and compensation  settings to analyse flow cytometric data on Tetramer+ and CD8+ cells? The small population of double positive cells in figure 3B seems different that the double positive cell population shown in figure 3A. How could this be explained?

8. Figure 5: Do NAC1-/- mice express less memory cells in the CD8+ population in general? If NAC1 would restrain memory formation, wouldn't it be expected that these mice have generated less memory cells to many of the antigens they have been exposed to in general until the time of assay?The plots of one specimen may suggest so. What is the overall picture when all specimens of the study would be analysed?

9. Figure 6 shows tissue-resident cells, but should show effector memory CD8+ cells and central memory cells accordingly to provide insight to the reader on the types of memory cells that result from NAC1-/-. Was there a statistically significant difference for these types of memory cells?

10. CHIP analysis shown in figure 7C seems to be done with cells that had not been stimulated with anti-CD3/28. How does this relate to analyses in figure 7A and 7B, where cells had been stimulated with anti-CD3/CD28? Shouldn't the CHIP analyses have been done on anti-CD3/CD28-stimulated cells accordingly?

11. What was the housekeeping gene that was used to normalize irf4 mRNA analyses by qPCR? Provide sequences of the primers used for the qPCR of this housekeeping gene in the methods section.

12. Describe statistical testing in the methods section.

13. Describe flow cytometry of memory T cells in the methods section.

14. Describe CFSE-labeling in the methods section.

15. Describe types of machines that were used for qPCR and flowcytometry.